# Incretins as a Potential Treatment Option for Gestational Diabetes Mellitus

**DOI:** 10.3390/ijms231710101

**Published:** 2022-09-03

**Authors:** Aleksandra Pilszyk, Magdalena Niebrzydowska, Zuzanna Pilszyk, Magdalena Wierzchowska-Opoka, Żaneta Kimber-Trojnar

**Affiliations:** Chair and Department of Obstetrics and Perinatology, Medical University of Lublin, 20-090 Lublin, Poland

**Keywords:** dipeptidyl peptidase-4, gestational diabetes mellitus, glucagon-like peptide-1, gliptins, incretins, liraglutide, pregnancy, sitagliptin, vildagliptin

## Abstract

Gestational diabetes mellitus (GDM) is a metabolic disease affecting an increasing number of pregnant women around the world. It is not only associated with numerous perinatal complications but also has long-term consequences impacting maternal health and fetal development. To prevent them, it is important to keep glucose levels under control. As much as 15–30% of GDM patients will require treatment with insulin, metformin, or glyburide. With that in mind, it is crucial to keep searching for novel and improved pharmacotherapies. Nowadays, there are ongoing studies investigating the use of other groups of drugs that have proven successful in the treatment of T2DM. Glucagon-like peptide-1 (GLP-1) receptor agonist and dipeptidyl peptidase-4 (DPP-4) inhibitor are among the drugs targeting the incretin system and are currently receiving significant attention. The aim of our review is to demonstrate the potential of these medications in treating GDM and preventing its later complications. It seems that both groups may be successful in the GDM management used alone or as an addition to better-known drugs, including metformin and glyburide. However, more clinical trials are needed to confirm their importance in GDM treatment and to demonstrate effective therapeutic strategies.

## 1. Introduction

Gestational diabetes mellitus (GDM) is a disorder defined as carbohydrate intolerance that manifests during the second or third trimester of pregnancy. It can completely resolve after delivery or persist as diabetes is revealed during pregnancy [1]. It is estimated that GDM affects 13% of pregnant women worldwide, and this percentage is expected to increase [2].

The occurrence of GDM affects the health of mothers and their offspring [3]. The perinatal complications associated with GDM are pre-term labor, preeclampsia, and shoulder dystocia [4,5]. A large percentage of such pregnancies require delivery by cesarean section [6]. Furthermore, complications of GDM have long-term effects. Women with GDM are at greater risk of type 2 diabetes mellitus (T2DM), and they also present a 63% higher risk of cardiovascular disease (CVD) [5,7].

Disorders of carbohydrate metabolism affect the fetus during the prenatal period but can also have long-term effects on the offspring [8]. Among the short-term consequences, the most common is fetal hypertrophy, which results in macrosomia at birth, but also insulin resistance that develops in the prenatal period, and postnatal hypoglycemia, which can be life-threatening for the child [9]. Children born to GDM mothers are at increased risk of developing T2DM, CVD, obesity, and other metabolic diseases later in life [1,4,10].

The exact mechanisms of GDM still remain unclear. However, many researchers point to two main etiological factors, pancreatic β-cell dysfunction and insulin resistance, secondary to placental hormonal release. Moreover, there are risk factors that significantly affect the incidence of GDM. They include a family history of diabetes, advanced maternal age, excessive weight or obesity, Westernized diet, and unhealthy lifestyle [1,4].

During a physiological pregnancy, multiple metabolic changes occur [11,12,13]. Insulin resistance develops from about the middle of pregnancy to the end of the third trimester. This process is caused by the increased levels of hormones and adipokines secreted from the placenta, including tumor necrosis factor (TNF)-α, human placental lactogen, and human placental growth hormone. In addition, changes in estrogen, progesterone, and cortisol concentrations are also thought to affect the glucose–insulin imbalance. The mother compensates for these changes, increasing insulin secretion by 200–250% [14,15].

Risk factors, especially obesity, affect the balance between hormone-induced insulin resistance and increased insulin secretion, which consequently results in GDM [16,17].

Another etiological factor of GDM is the dysfunction of pancreatic β-cells. Hyperglycemia, generated as a result of insulin resistance, stimulates β-cells to secrete more insulin. However, if the demand for insulin is too high, the phenomenon of glucotoxicity occurs, resulting in the overburdening and dysfunction of the β-cells that leads to hyperglycemia [1,16].

Molecules that also affect the insulin–glucose balance are incretin hormones such as glucose-dependent insulinotropic polypeptide (GIP) and glucagon-like peptide-1 (GLP-1). Incretin peptides are of great importance in glucose homeostasis as they increase insulin secretion by β-cells, reduce insulin resistance as well as inhibit β-cell apoptosis and stimulate β-cell proliferation [18,19].

In addition, oxidative stress and chronic inflammation have an impact on the development of GDM [1,14,19]. An imbalance between pro- and antioxidants can lead to damage to the cells responsible for the glucose–insulin balance [1]. Moreover, the expression of pro-inflammatory cytokines in chronic inflammation contributes to increased insulin resistance, but the mechanism underlying this phenomenon remains unclear and requires further study [19].

## 2. Management of GDM

The goal of GDM therapy is to achieve and maintain glycemia at levels found in healthy women. To achieve it, a multidisciplinary approach is required, consisting of dietary modifications, lifestyle changes, physical activity, and monitoring of blood glucose levels [20,21].

Lende et al. indicated that 70–85% of patients with GDM can be managed due to the aforementioned changes without the use of pharmacotherapy [21]. The general recommendation for pregnant women is to consume three meals and two snacks a day. Diet plans for pregnant patients with diabetes should have a similar layout, but the details should be discussed with a nutrition educator. Physical activity guidelines recommend 30 min of moderate intensity exercise, 5 days a week [10,21].

Unfortunately, despite following the recommendations discussed above, as much as 15–30% of patients with GDM will require pharmacotherapy. The treatment of hyperglycemia in patients with GDM includes insulin, as well as oral medications, including metformin and glyburide [10,21,22,23,24]. Due to its structure, insulin is considered a safer drug than metformin and glyburide [25]. It is a larger molecule and thus does not pass through the placenta, unlike metformin and glyburide. Moreover, Lende et al. show the superiority of insulin due to the ability to personalize treatment and adjust the dose and timing of its administration to the patient’s blood glucose levels [21].

Metformin and glyburide are under ongoing study, but the results are not sufficiently robust to draw a firm conclusion. According to the study cited by Johns et al., there are currently insufficient data to compare the advantages of any particular oral therapy over another. Moreover, the long-term effects of intrauterine exposure to metformin and glyburide are unknown and require further study [20].

Molecules that have attracted great interest among researchers are incretin peptides, which are used in the treatment of T2DM. Studies show that reduced levels of incretin peptides such as GLP-1 can affect the development of GDM [19]. Moreover, Fritsche et al. indicate that higher levels of GLP-1 may protect against fetal hypertrophy [2]. However, researchers agree that further studies and research are needed to translate these observations into therapeutic strategies [2,19].

The purpose of the following paper is to review the literature on incretin peptides as potential drugs for GDM. Although these molecules raise high hopes among researchers, many scientific papers present contradictory results. For this reason, it is necessary to collect the research results obtained so far and put them under analysis. Our hope is that the conclusions drawn regarding the effects of incretins on hyperglycemia, as well as maternal and fetal well-being, will contribute to the development of potential treatments for GDM.

## 3. GLP-1—State of the Art

GLP-1 is a peptide that is produced through the proteolysis of proglucagon, a protein expressed in L cells in the intestinal mucosa, α cells of the pancreas, as well as in the nucleus of the solitary tract (NTS) in the brainstem. GLP-1 has access to a specific GLP-1 receptor (GLP-1R) that is expressed in a wide range of target tissues. It is secreted mainly after the ingestion of glucose, lipids, or mixed meals, and increases glucose-stimulated insulin secretion at physiological plasma concentrations, which meets all the criteria for an incretin hormone [26,27].

GLP-1 protects against hyperglycemia by inhibiting glucagon secretion from pancreatic α cells, complementing its effect to stimulate insulin, and coordinating the islet hormones leading to lower blood glucose. Notably, GLP-1 increases insulin release only in the context of hyperglycemia and therefore does not cause hypoglycemia. In addition, GLP-1 actions include delay in gastric emptying and small intestinal transit induced. GLP-1 and GLP-1R agonists induce an increase in β-cell mass through enhanced cellular regeneration and the inhibition of apoptosis [27,28,29,30].

The GLP-1 molecule is activated by the action of proconvertase 1, which results in the formation of two bioactive forms of GLP-1, GLP-1 (7–36) and GLP-1 (7–37). The inactivation of them proceeds by the cleaving of two amino acids at its N-terminus by the enzyme dipeptidyl peptidase-4 (DPP-4). The inactivation is rapid, rendering the GLP-1 half-life a mere 2–3 min [31]. In order to improve the clinical efficacy to T2DM and obesity, long-lasting analogs for GLP-1 have been developed. Their function is to delay metabolism by DPP-4, and they extend their half-life in the blood [32]. GLP-1R agonists can be formed from the active fragment of the human GLP-1 (7–36) or derived from exendin-4. Exendin-4 was isolated from the saliva of the Gila monster (Heloderma suspectum), and since the DPP-4 enzymatic cleavage site is not present in this peptide, it is resistant to degradation in serum, resulting in an increased biological half-life. Synthetic exendin-4 was named exenatide and was the first GLP-1R approved to treat T2DM [33,34].

When analyzing the possible mechanisms of incretin’s effects in GDM, the relationship with substance P should also be considered. Substance P, as an insulin secretagogue peptide, may play a role in the pathogenesis of T2DM and GDM [35]. Substance P sensory cells co-cultured with sensory neurons showed both the GLP-1R agonist exendin-4 and the GIP-induced transient increase in calcium directly coupled to substance P secretion in sensory nerves. These data suggest the actions of GLP-1 and GIP as neurolymphocrines and explain the mechanism by which the sensory nerves respond to the postprandial state through the mesenteric lymphatic fluid [36].

Based on the first 10 years of GLP-1 therapeutics, the use of GLP-1R agonists for the treatment of T2DM and obesity is growing. They induce significant improvements in glycemic control and weight loss, combined with cardioprotection in individuals at risk of or with pre-existing CVD [37]. Therefore, they are increasingly being used in new-generation therapies showing growing effectiveness while maintaining safety in the treatment of diabetes, obesity, non-alcoholic steatohepatitis, and related cardiometabolic disorders. Perhaps GLP-1R therapy is also effective in improving insulin action and reducing body weight in overweight patients with GDM.

## 4. GLP-1 and Pregnancy

Valsamakis et al. investigated the possible physiological associations of gut hormone levels (including GLP-1) in a group of non-obese, non-diabetic pregnant women during three trimesters of pregnancy with maternal glucose homeostasis, body weight, and fetal growth [38]. Fasting GLP-1 levels increased from the second to the third trimester and correlated negatively with fetal abdominal circumference, birth weight, and maternal insulin secretion. The authors revealed that GLP-1 levels in the first trimester were the best negative predictors of fetal abdominal circumference in the second trimester and maternal weight change during pregnancy. Their results indicate that during physiological pregnancy, maternal GLP-1 may be involved in mechanisms that compensate for pregnancy-related increases in glycemia and insulin resistance, suggesting a role for this peptide in maternal metabolism and body weight, as well as in fetal growth [38].

It is proven that GDM poses a higher risk of developing T2DM in the future; both conditions have a similar spectrum of metabolic changes [39]. The incretin effect is reduced in T2DM, which may be due to decreased secretion of the incretin hormones or reduced response of the pancreatic β-cells to them. The role of GLP-1 in the pathogenesis of GDM is still unclear.

Bonde et al. investigated the GLP-1 response in pregnant women with and without GDM and again after the delivery when normal glucose tolerance (NGT) was restored [40]. The authors observed that pregnancy was associated with a low postprandial GLP-1 response. The patients with GDM had decreased postprandial GLP-1 levels during pregnancy compared to the 3–4 months postpartum period. Bonde et al. suggested that patients with GDM are characterized by impaired postprandial incretin hormonal responses with decreased GLP-1 activity during pregnancy that normalizes postpartum [40]. The reversibility of the reduced postprandial GLP-1 response in GDM patients may suggest that it develops secondary to insulin resistance or diabetes and is not the major pathogenetic defect in the development of diabetes [40].

In addition, Kosinski et al. indicate that women with GDM show a reduced incretin effect, which is completely reversible with the restoration of normal glucose homeostasis [41]. Mosavat et al. found reduced GLP-1 levels during pregnancy in patients with GDM compared to a control group [19]. Confirming previous studies, GLP-1 levels in the GDM group showed a decrease during pregnancy and immediately after delivery and then increased in the late post-puerperium [19]. Based on current knowledge, a decrease in incretin levels might represent the mechanisms that compensate for the tendency to pregnancy-related increase in glycemia and insulin resistance. Decreased GLP-1 function in GDM may be an early abnormality that may signify the need for appropriate monitoring and treatment.

## 5. GLP-1R Agonists in GDM

According to several clinical trials, metformin and glyburide are drugs with acceptable safety that can be used for GDM pharmacotherapy. However, it is necessary to continuously search for new medications. We reviewed medical publications that reported on the use of GLP-1R agonists in the treatment of GDM.

Elkind-Hirsch et al. evaluated the additive effect of treatment with the long-acting GLP-1 analog, liraglutide, in patients with postpartum GDM [42]. Liraglutide is a GLP-1R agonist used for the treatment of T2DM and obesity. They compared whether drug treatment with liraglutide added to metformin therapy is more effective than metformin alone in improving metabolic and anthropometric parameters in a population of overweight women with a history of GDM. It was a double-blinded randomized controlled study comparing two treatment groups for 80–84 weeks. Their study shows that liraglutide attached to metformin with high efficacy improves insulin action, weight loss, reduces abdominal obesity, reduces mean glucose levels in the oral glucose tolerance test, and improves lipid profile, compared to metformin used alone in the treatment of postpartum overweight patients with GDM [42].

It has been shown that there is an association with the incidence of non-alcoholic fatty liver disease (NAFLD) in relatively young and not severely obese women with prior GDM [43]. To date, there are known beneficial effects of GLP-1R agonist treatment in patients with NAFLD. Vedtofte et al. [44] examined the effect of liraglutide in preventing the development of NAFLD or resolving NAFLD in a population of overweight or obese women with a history of GDM but without current diabetes. After one year of liraglutide treatment, they received the following results: liraglutide had no effect on the presence of ultrasound-diagnosed NAFLD in overweight or obese non-diabetic women with prior GDM but reduced body weight, fasting plasma glucose (FPG), and hemoglobin A1c (HbA1c) compared to placebo treatment [44]. One of the most important therapeutic goals in GDM is blood glucose control. Foghsgaard et al. reported that blood glucose levels, HbA1c, and the risk of developing carbohydrate disorders could be reduced after treatment with liraglutide [45]. However, they repeated the study after a week of wash-out and found that the normalization of FPG and glucose tolerance was not maintained. In contrast, improvements in HbA1c were sustained after the wash-out [45]. Noteworthy, the investigators in the other studies rarely reported that they would repeat the study after a period of treatment interruption. This represents an important aspect in the continuation of treatment. The most important question is whether this treatment benefits patients, and therefore further studies are needed.

The aforementioned studies differ in the size of the study group, the population risk of diabetes, and the diagnostic criteria for GDM. Vedtofte et al. [44] and Foghsgaard et al. [45] diagnosed GDM according to the cutoff point found in the Danish guidelines, which is a plasma glucose level ≥ 9.0 mmol/L 2 h after 75 g-oral glucose tolerance test (OGTT). Elkind-Hirsch [42] included the criteria according to the American Diabetes Association. Mean blood glucose concentrations during the OGTT were calculated by summing the glucose values obtained at 0, 30, 60, and 120 min and dividing them by 4.

This systematic review suggests that GLP-1R agonists have beneficial effects in patients with GDM, but the current data cannot prove their use in GDM. Additional clinical trials are required.

## 6. DPP-4—State of the Art

The DPP family consists of enzymes such as DPP-4, fibroblast activation protein α (FAP; seprase), DPP-8, DPP-9, and prolyl carboxypeptidase (PCP; angiotensinase C) [46]. DPP-4, also known as T-cell differentiation antigen CD26, is a serine protease present in the circulation as two isoforms—a soluble protein and plasma membrane-bound [47,48]. The soluble isoform enables intercellular contact by cleaving protein substrates in body fluids, while the plasma membrane-bound affects intercellular communication through receptor activity [48].

Human DPP-4/CD26 has a short intracellular domain (6 amino acids), a transmembrane region, and an extracellular domain that possesses DPP activity [46]. DPP-4 expression was observed in numerous cells such as intestinal K cells, hepatocytes, adipocytes, renal brush border membranes, bone marrow cells, pancreatic cells, placental cytotrophoblasts, endothelial cells, and on the surface of lymphocytes [46,48,49,50,51].

According to the research, DPP-4 plays a role in regulating metabolism, appetite, energy expenditure, and body composition [48]. The abnormal expression and activity of DPP-4 has been observed to be associated with the occurrence of hyperglycemia and an increase in body mass index, implying that this enzyme may contribute to the development of diseases such as T2DM and obesity by mediating inflammation and insulin resistance in adipose tissue [48]. Some of the other functions of DPP-4 include interactions with various proteins such as streptokinase and plasminogen, extracellular matrix components (collagen, fibronectin), and the role of binding sites for the chemokine CXCR4 receptor. In addition, sCD26 has the potential to enhance the congenital immune response [50]. Due to the diverse functions of DPP-4, the dysfunction of this molecule has additional implications for the onset of inflammatory, viral entry, and immune-mediated diseases [47,52].

However, DPP-4 is most known for the mediation of GLP-1 and GIP inactivation. GLP-1 and GIP, through their effects on pancreatic β-cells, are responsible for stimulating insulin secretion [53]. In addition, by affecting α-cells, they mediate the inhibition of glucagon release and thus reduce hepatic glucose production [49]. Through repression of the activity of DPP-4, it is possible to prevent the degradation of GLP-1 and GIP, thereby increasing their concentrations by 2–3 times and leading to an increase in postprandial glucose-dependent insulin concentration and a decrease in glucagon release [54,55]. For this mechanism to be effective, having preserved endogenous incretin production is required [55]. DPP-4 inhibition also appears to affect insulin sensitivity through non-enzymatic interactions with membrane proteins such as caveolin-1 [47]. In conclusion, DPP-4 inhibition improves pancreatic β-cell proliferation by inhibiting apoptosis pathways and thus mediates improved glucose homeostasis without inducing hypoglycemia [53,56].

The group of DPP-4 inhibitors includes vildagliptin, sitagliptin, saxagliptin, linagliptin, and alogliptin [57]. Their activity is based on stimulating insulin secretion in a glucose-dependent mechanism, and compared to other groups of antidiabetic drugs, demonstrates a less rapid glucose-lowering effect. Moreover, gliptins are weight neutral due to a limited increase in GLP-1 activity [53]. These drugs are primarily used in adults with T2DM in monotherapy in case of contraindications or lack of efficacy of metformin treatment and in combination with other antidiabetic drugs, including insulin [51]. All approved DPP-4 inhibitors appear to have similar glycemic effectiveness resulting in a reduction in HbA1c; however they provide fewer advantages compared to GLP-1R agonists and sulfonylureas [51,55].

In addition to their evident glucose-lowering effects, DPP-4 inhibitors exhibit anti-inflammatory activity, which, according to the research, controls vascular aging [49]. Furthermore, they show nephroprotective effects; however, the mechanisms have not been documented so far. Saxagliptin seems to be the most effective since, according to studies, it can inhibit the progression of microalbuminuria [58]. On the other hand, alogliptin and sitagliptin show protective effects on the cardiovascular system [59].

## 7. DPP-4 in GDM

The role of DPP-4 activity in the pathogenesis of GDM is not fully understood. Its significance in fetal development is also under investigation. An important issue in this context is the concept of fetal programming. Montaniel et al. [48] point out that the children of mothers who are obese during pregnancy are more likely to develop obesity during their lives, but the mechanisms behind this correlation are still unknown. The researchers suggest that plasma DPP-4 activity, which is elevated in the male offspring of mothers with obesity, may play an essential role here. Montaniel et al. used a mouse model of maternal high-fat diet-induced obesity and sitagliptin to assess whether it can inhibit DPP-4 activity in vivo and, thus, block fetal programming toward obesity. This analysis proved that sitagliptin is effective only on male offspring, without an effect on female ones. They indicate that DPP-4 inhibitor therapy during pregnancy could inhibit fetal programming toward obesity in the offspring of obese mothers, but more detailed research is required to confirm this observation [48].

Al-Aissa et al. [60] conducted a study to evaluate DPP-4 activity in the cord blood samples from 270 patients—111 with GDM and 159 controls. The GDM patients received effective diet or insulin treatment. As a result, they demonstrated lower DPP-4 enzymatic activity in newborns of mothers with GDM compared to healthy controls. They believe it may represent a regulatory mechanism involved in fetal programming, which may affect the development of metabolic diseases in these offspring in the future [60].

Liu et al. [61], to assess the impact of DPP-4 on the pathogenesis of GDM, decided to perform research in order to compare the values of this molecule in maternal and umbilical cord serum in pregnant GDM patients and healthy controls. Insulin values decrease during pregnancy in women with GDM. The researchers hypothesized that DPP-4 concentrations might be higher in these patients due to its ability to lower insulin via the degradation of incretins. A strong correlation was found between maternal and umbilical cord venous DPP-4 levels. However, no significant difference was presented in maternal or umbilical cord venous DPP-4 concentrations when comparing the GDM and control group. In addition, the researchers were interested in evaluating the molecule’s correlation with neonatal birth anthropometry; however, no association was documented. The investigators believe that DPP-4 probably does not play a significant role in the pathogenesis of GDM and has no effect on fetal development [61].

Kandzija et al. [62] conducted a study to investigate the activity and concentrations of DPP-4 associated with syncytiotrophoblast-derived extracellular vesicles (STB-EVs) in patients with pregnancies complicated by GDM. A syncytiotrophoblast is responsible for releasing extracellular vesicles (EVs) into the maternal circulation to mediate intercellular communication between the placenta and maternal metabolism. The concentration of STB-EVs increases with the duration of pregnancy and positively correlates with the onset of insulin resistance during gestation. The researchers documented an increase in the concentration of circulating small STB-EVs in maternal plasma in pregnant patients complicated by GDM compared to healthy controls. In addition, they also noted that STB-EVs contain active enzyme DPP-4 capable of degrading GLP-1-DPP-4 values, which were more than eight times higher in women with GDM. However, they note the limitations of their analysis, including the small sample size (n = 6 for both GDM and controls), and emphasize the need for further studies on a larger scale to clarify the roles of DPP-4 associated with STB-EV [62].

Although DPP-4 inhibitors are widely used among adult T2DM patients, the number of studies evaluating their use in women with pregnancies complicated with GDM and during the postpartum period is still limited.

Sun et al. [63] evaluated the ability of sitagliptin to reduce insulin resistance and alleviate GDM symptoms. They examined 206 patients diagnosed with GDM during the second trimester of pregnancy, 102 of them were treated with sitagliptin, while 104 women received a placebo. After 16 weeks of the study, patients in the sitagliptin group achieved significant reductions in FPG and serum insulin levels over values of the above parameters obtained at the beginning of the study, compared to patients in the placebo group. In addition, they showed that the use of sitagliptin helped down-regulate retinol-binding protein-4 (RBP-4), a molecule that is known to be positively correlated with the severity of glucose intolerance and insulin resistance in women with previous GDM. Among patients receiving sitagliptin and their newborns, no adverse effects were observed during the follow-up [63].

Hummel et al. [64] assessed the effectiveness of vildagliptin for the prevention of postpartum diabetes in women with a recent history of insulin-requiring GDM. The study included 113 women—58 received vildagliptin and 55 placebo pills. The patients received 50 mg of vildagliptin twice a day for 24 months, starting therapy on average about 9 months after delivery, and were additionally monitored for 12 months after completing the treatment. However, the researchers were unable to demonstrate an advantage of taking vildagliptin over a placebo in terms of reducing the risk of postpartum diabetes. Moreover, indicators related to insulin resistance, glucose control, or pancreatic β-cell function were not significantly different between the two groups [64].

In turn, Elkind-Hirsch et al. [65] proved that combined therapy consisting of sitagliptin and metformin was more effective in women with impaired postpartum glucose regulation compared to metformin alone or placebo in women with a history of GDM.

The presented studies differ in the size of the study group, the population risk of diabetes, as well as in the diagnostic criteria for GDM. The criteria of 75 g OGTT (according to IADPSG) were included in the studies conducted by Liu et al. [61], Kandzija et al. [62], and Sun et al. [63], while the criteria of 100 g OGTT (according to Carpenter–Coustan) in the study performed by Elkind-Hirsch et al. [65]. The research by Al-Aissa et al. [60] took place in two countries that used different diagnostic criteria for GDM: the Austrian group according to the IADPSG criteria, while the Hungarian group followed the modified 1999 WHO recommendation, namely-75 g OGTT at 24–28 gestational weeks, where FPG ≥ 6.1 mmol/L or 2-h plasma glucose ≥ 7.8 mmol/L qualified for a diagnosis of GDM. Hummel et al. [64] classified women as having GDM if two of three capillary blood glucose values measured during a 75 g OGTT were: FPG >5 mmol/L, 1-h plasma glucose >10.6 mmol/L or 2-h plasma glucose >8.9 mmol/L.

There are some mixed conclusions from the abovementioned studies. Undoubtedly, further research on the use of DPP-4 inhibitors in the treatment of GDM and their possible impact on reducing fetal programming for obesity and metabolic diseases.

## 8. Potential Side Effects of the Use of Incretins in Pregnant Women and Their Offspring

Despite the many positive aspects of using incretins in pregnant patients with diabetes, it is important to consider the potential adverse effects of incretin-based therapies. Due to the small amount of data, we can only consider potential symptoms that have been discussed in the available published literature.

The most commonly reported side effects of incretin drugs include nausea and diarrhea. It can be speculated that the use of these drugs may exacerbate nausea in the first trimester of pregnancy [66]. Nevertheless, a second-generation GLP-1R agonists conjugate composed of vitamin B12 and exendin-4 was reported to reduce the adverse reactions, such as nausea and vomiting, exhibiting improved tolerance [67,68].

The use of incretin drugs can also affect the developing fetus. According to Lende et al., this type of therapy can lead to fetal hypertrophy, which may cause a number of perinatal complications, as well as neonatal hypoglycemia [21].

One of the potential risks of using incretins is the development of pancreatic cancer and pancreatitis. In their paper, Chia et al. discuss a 2005 study according to which subjects treated with GLP-1R agonists had a 10-fold increase in the incidence of pancreatitis and a 2.9-fold increase in pancreatic cancer. However, these results remain inconsistent with a review of preclinical and retrospective studies conducted by the European Medicines Agency (EMA) and the United States of America (USA) Food and Drug Administration (FDA). Not only did the researchers fail to confirm these results, but also, they found no increased incidence of the aforementioned diseases with the use of the drug in question [66].

It is very important to note that the listed side effects mainly apply to patients with T2DM. Their occurrence in women with GDM is only hypothetical and requires more thorough research and confirmation by studies.

## 9. Conclusions

According to numerous clinical studies, the GDM pharmacotherapy used so far is acceptably safe. However, it is necessary to constantly search for new and improved medications. An alternative option in the treatment of GDM may be incretin peptides, which raise high hopes among researchers.

Incretin peptides are of great importance in glucose homeostasis as they increase insulin secretion by β-cells, reduce insulin resistance, inhibit β-cell apoptosis, and stimulate β-cell proliferation. Thanks to these mechanisms, they can also bring potential benefits (listed in Figure 1) in the case of GDM.

Nevertheless, scientists agree that more research is needed to translate these observations into therapeutic strategies, especially to confirm the safety of the therapy not only for the mother but also for the fetus. This type of pharmacotherapy will be investigated in the coming years and may bring promising results, particularly for patients with obesity and excessive gestational weight gain. The increasing use of incretins in the treatment of diabetes and obesity will also provide data on their effects in early pregnancy when it occurs unplanned. However, drawing firm conclusions on such a correlation requires long-term observation of the patients’ offspring.

## Figures and Tables

**Figure 1 ijms-23-10101-f001:**
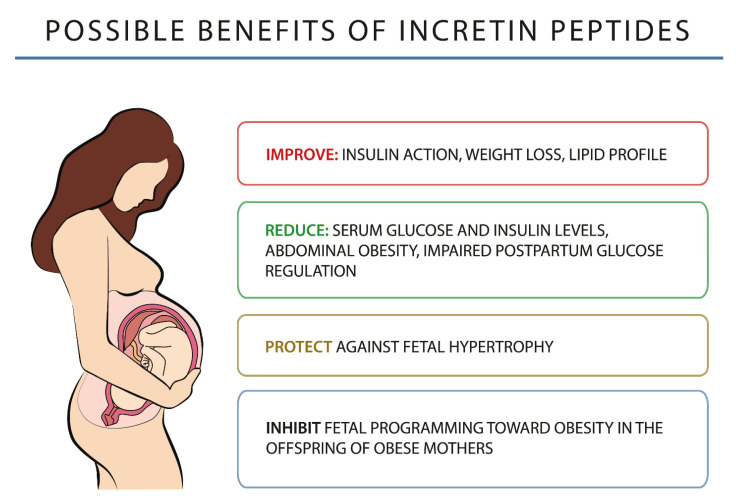
Possible benefits of incretin peptides.

## Data Availability

Not applicable.

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
