# Peer review of "Incretins as a Potential Treatment Option for Gestational Diabetes Mellitus"

_ijms, 2022, doi:10.3390/ijms231710101_

Round 1

Reviewer 1 Report

The authors presented a review on the feasibility of using GLP-1 agonist in women with gestational diabetes mellitus. Although the content is reasonable and acceptable, it tends to focus only on the positive aspects of GLP-1 agonist. As a matter of interest to the readers, authors should demonstrate the possible side effects of the use of GLP-1 in the pregnant state. Does treatment that results in GLP-1 concentrations several times the bioactive have no effect on the mother and fetus? GLP-1 agonist has been reported to contribute to the development of thyroid carcinoma and the development of pancreatitis. I think authors should present animal or human data in this regard. In addition, certain GLP-1 agonist also potently induces nausea and weight loss. Does it work badly against for pregnant women?

Furthermore, there are the following minor point to be revised

Line 6073

The following two representations are mixed: β cells/β-cells

Line 155

It would be better to note in more detail the contents of dedication Reference 38, especially with regard to the results. It is difficult to understand what results in the literature in Reference 38 indicate "GLP-1 compensates for insulin resistance in pregnancy".

Line 168

The same problem as in Reference 38 applies with respect to the citation in Reference 40.

Author Response

Dear Reviewer 1 

            We would like to resubmit our manuscript (ID manuscript: ijms-1901881 entitled “Incretins as a Potential Treatment Option for Gestational Diabetes Mellitus”. We appreciate your valuable remarks and hope that the quality of our manuscript is going to meet your expectations now that we have made some suggested alternations.

               We have rearranged our paper in accordance with your valuable comments and suggestions. The manuscript has been checked by a native-speaker.  

“The authors presented a review on the feasibility of using GLP-1 agonist in women with gestational diabetes mellitus. Although the content is reasonable and acceptable, it tends to focus only on the positive aspects of GLP-1 agonist. As a matter of interest to the readers, authors should demonstrate the possible side effects of the use of GLP-1 in the pregnant state. Does treatment that results in GLP-1 concentrations several times the bioactive have no effect on the mother and fetus? GLP-1 agonist has been reported to contribute to the development of thyroid carcinoma and the development of pancreatitis. I think authors should present animal or human data in this regard. In addition, certain GLP-1 agonist also potently induces nausea and weight loss. Does it work badly against for pregnant women?

Thank you very much for your perceptive comments. Agreeing with your opinion, we have included the following subsection:

  1. Potential side effects of the use of incretins in pregnant women and their offspring

Despite the many positive aspects of using incretins in pregnant patients with diabetes, it is important to consider the potential adverse effects of incretin-based therapies. Due to the small amount of data, we can only consider potential symptoms that have been discussed in the available published literature.

The most commonly reported side effects with incretin drugs include nausea and diarrhea. It can be speculated that the use of these drugs may exacerbate nausea in the first trimester of pregnancy [66]. Nevertheless, a second-generation GLP-1R agonists conjugate composed of vitamin B12 and exendin-4 was reported to reduce the adverse reactions, such as nausea and vomiting, exhibiting improved tolerance [67,68].

The use of incretin drugs can also affect the developing fetus. According to Lende et al., this type of therapy can lead to fetal hypertrophy, which may cause a number of perinatal complications, as well as neonatal hypoglycemia [21].

One of the potential risks of using incretins is the development of pancreatic cancer and pancreatitis. In their paper, Chia et al. in discuss a 2005 study, according to which subjects treated with GLP-1R agonists had a 10-fold increase in the incidence of pancreatitis and a 2.9-fold increase in pancreatic cancer. However, these results remain inconsistent with a review of preclinical and retrospective studies conducted by the European Medicines Agency (EMA) and the United States of America (USA) Food and Drug Administration (FDA). Not only did the researchers fail to confirm these results, but also they found no increased incidence of the aforementioned diseases with the use of the drug in question [66].

It is very important to note that the listed side effects mainly apply to patients with T2DM. Their occurrence in women with GDM is only hypothetical and requires more thorough research and confirmation by studies.

“Furthermore, there are the following minor point to be revised

Line 6073

The following two representations are mixed: β cells/β-cells”

Thank you very much. They have been corrected.

“Line 155

It would be better to note in more detail the contents of dedication Reference 38, especially with regard to the results. It is difficult to understand what results in the literature in Reference 38 indicate "GLP-1 compensates for insulin resistance in pregnancy".

According to your comments we have changed this part as follows:

Valsamakis et al. investigated the possible physiological associations of gut hormone levels (including GLP-1) in a group of non-obese, non-diabetic pregnant women during three trimesters of pregnancy with maternal glucose homeostasis, body weight, and fetal growth [38]. Fasting GLP-1 levels increased from the second to third trimester and correlated negatively with fetal abdominal circumference, birth weight, and maternal insulin secretion. The authors revealed that GLP-1 levels in the first trimester were the best negative predictors of fetal abdominal circumference in the second trimester and maternal weight change during pregnancy. Their results indicate that during physiological pregnancy, maternal GLP-1 may be involved in mechanisms that compensate for pregnancy-related increases in glycemia and insulin resistance, suggesting a role for this peptide in maternal metabolism and body weight, as well as in fetal growth [38].

Line 168

The same problem as in Reference 38 applies with respect to the citation in Reference 40.

Thank you for your advice, we have also changed this part as follows: 

Bonde et al. investigated the GLP-1 response in pregnant women with and without GDM and again after the delivery, when normal glucose tolerance (NGT) was restored [40]. The authors observed that pregnancy was associated with a low postprandial GLP-1 response. Patients with GDM had decreased postprandial GLP-1 levels during pregnancy compared to the 3-4 months postpartum period. Bonde et al. suggested that patients with GDM are characterized by impaired postprandial incretin hormonal responses with decreased GLP-1 activity during pregnancy that normalizes postpartum [40]. The reversibility of the reduced postprandial GLP-1 response in GDM patients may suggest that it develops secondary to insulin resistance or diabetes, and is not the major pathogenetic defect in the development of diabetes [40].

              We would like to take this opportunity to thank the Reviewer for all the valuable and highly perceptive remarks which have definitely made a substantial contribution to the quality of our paper. Thank you for considering our manuscript for publication. We appreciate your time and look forward to hearing from you.

Yours faithfully,

Prof. Zaneta Kimber-Trojnar, MD PhD

Department of Obstetrics and Perinatology

Medical University of Lublin

Jaczewskiego 8, 20-954 Lublin, Poland

Tel: +48 81 7244 769; Fax: +48 81 7244 841

E-mail: zkimber@poczta.onet.pl

Reviewer 2 Report

In this manuscript, the authors review the potential of incretins in treating GDM and preventing its later complications. The manuscript is overall very detailed and well written. However, I recommend adding the following details to make this manuscript having a more balanced view:

1. Current known side effects of incretins.

2. The potential side effects of incretins which may affect pregnancy and fetus development (this is briefly mentioned in the conclusion, but I think need some elaboration and hypothesis for further studies).

Author Response

Dear Reviewer 2 

            We would like to resubmit our manuscript (ID manuscript: ijms-1901881 entitled “Incretins as a Potential Treatment Option for Gestational Diabetes Mellitus”. We appreciate your valuable remarks and hope that the quality of our manuscript is going to meet your expectations now that we have made some suggested alternations.

We have rearranged our paper in accordance with your valuable comments and suggestions. The manuscript has been checked by a native-speaker.

“In this manuscript, the authors review the potential of incretins in treating GDM and preventing its later complications. The manuscript is overall very detailed and well written. However, I recommend adding the following details to make this manuscript having a more balanced view:

  1. Current known side effects of incretins.
  2. The potential side effects of incretins which may affect pregnancy and fetus development (this is briefly mentioned in the conclusion, but I think need some elaboration and hypothesis for further studies).

Thank you very much for your perceptive comments. Agreeing with your opinion, we have included the following subsection:

  1. Potential side effects of the use of incretins in pregnant women and their offspring

Despite the many positive aspects of using incretins in pregnant patients with diabetes, it is important to consider the potential adverse effects of incretin-based therapies. Due to the small amount of data, we can only consider potential symptoms that have been discussed in the available published literature.

The most commonly reported side effects with incretin drugs include nausea and diarrhea. It can be speculated that the use of these drugs may exacerbate nausea in the first trimester of pregnancy [66]. Nevertheless, a second-generation GLP-1R agonists conjugate composed of vitamin B12 and exendin-4 was reported to reduce the adverse reactions, such as nausea and vomiting, exhibiting improved tolerance [67,68].

The use of incretin drugs can also affect the developing fetus. According to Lende et al., this type of therapy can lead to fetal hypertrophy, which may cause a number of perinatal complications, as well as neonatal hypoglycemia [21].

One of the potential risks of using incretins is the development of pancreatic cancer and pancreatitis. In their paper, Chia et al. in discuss a 2005 study, according to which subjects treated with GLP-1R agonists had a 10-fold increase in the incidence of pancreatitis and a 2.9-fold increase in pancreatic cancer. However, these results remain inconsistent with a review of preclinical and retrospective studies conducted by the European Medicines Agency (EMA) and the United States of America (USA) Food and Drug Administration (FDA). Not only did the researchers fail to confirm these results, but also they found no increased incidence of the aforementioned diseases with the use of the drug in question [66].

It is very important to note that the listed side effects mainly apply to patients with T2DM. Their occurrence in women with GDM is only hypothetical and requires more thorough research and confirmation by studies.

              We would like to take this opportunity to thank the Reviewer for all the valuable and highly perceptive remarks which have definitely made a substantial contribution to the quality of our paper. Thank you for considering our manuscript for publication. We appreciate your time and look forward to hearing from you.

Yours faithfully,

Prof. Zaneta Kimber-Trojnar, MD PhD

Department of Obstetrics and Perinatology

Medical University of Lublin

Jaczewskiego 8, 20-954 Lublin, Poland

Tel: +48 81 7244 769; Fax: +48 81 7244 841

E-mail: zkimber@poczta.onet.pl

Reviewer 3 Report

In this study, the authors reported  a potential treatment option for patients with gestational diabetes mellitus. The necessary and sufficient literature has been reviewed, and the paper is concise. However, there are some parts of the paper that can be add points of discussion.

Major points

1. The diagnostic criteria for gestational diabetes may differ from a study to another study. This point needs to be discussed.

2. Previous reports of adverse effects of dipeptidyl peptidase-4 (DPP-4) inhibitor in pregnant women and fetuses should also be reviewed. Comparison with the frequency of adverse reactions in the general population should also be considered, if appropriate.

3. In previous reports, is there any consideration of side effects in pregnant women for glucagon-like peptide-1 receptor agonist (GLP1RA) as well as the point made in 2. The GLP1 concentration during GLP1RA administration is much higher than the physiological concentration. With a view to the fact, it would be desirable to more carefully discuss the previously reported effects on pregnant women and the fetus.

Author Response

Dear Reviewer 3 

            We would like to resubmit our manuscript (ID manuscript: ijms-1901881 entitled “Incretins as a Potential Treatment Option for Gestational Diabetes Mellitus”. We appreciate your valuable remarks and hope that the quality of our manuscript is going to meet your expectations now that we have made some suggested alternations.

               We have rearranged our paper in accordance with your valuable comments and suggestions. The manuscript has been checked by a native-speaker. 

“In this study, the authors reported  a potential treatment option for patients with gestational diabetes mellitus. The necessary and sufficient literature has been reviewed, and the paper is concise. However, there are some parts of the paper that can be add points of discussion.

Major points

  1. The diagnostic criteria for gestational diabetes may differ from a study to another study. This point needs to be discussed.”

According to your comments we have changed these parts. 

  1. Previous reports of adverse effects of dipeptidyl peptidase-4 (DPP-4) inhibitor in pregnant women and fetuses should also be reviewed. Comparison with the frequency of adverse reactions in the general population should also be considered, if appropriate.
  2. In previous reports, is there any consideration of side effects in pregnant women for glucagon-like peptide-1 receptor agonist (GLP1RA) as well as the point made in 2. The GLP1 concentration during GLP1RA administration is much higher than the physiological concentration. With a view to the fact, it would be desirable to more carefully discuss the previously reported effects on pregnant women and the fetus.

Thank you very much for your perceptive comments. Agreeing with your opinion, we have included the following subsection entitled “8. Potential side effects of the use of incretins in pregnant women and their offspring.

              We would like to take this opportunity to thank the Reviewer for all the valuable and highly perceptive remarks which have definitely made a substantial contribution to the quality of our paper. Thank you for considering our manuscript for publication. We appreciate your time and look forward to hearing from you.

Yours faithfully,

Prof. Zaneta Kimber-Trojnar, MD PhD

Department of Obstetrics and Perinatology

Medical University of Lublin

Jaczewskiego 8, 20-954 Lublin, Poland

Tel: +48 81 7244 769; Fax: +48 81 7244 841

E-mail: zkimber@poczta.onet.pl

Round 2

Reviewer 1 Report

This article has been appropriately revised and there are no further comments.